# RETHINKING CURRICULUM LEARNING WITH INCREMENTAL LABELS AND ADAPTIVE COMPENSATION

## ABSTRACT

Like humans, deep networks learn better when samples are organized and introduced in a meaningful order or curriculum (Weinshall et al., 2018). While conventional approaches to curriculum learning emphasize the difficulty of samples as the core incremental strategy, it forces networks to learn from small subsets of data while introducing pre-computation overheads. In this work, we propose *Learning with Incremental Labels and Adaptive Compensation* (LILAC), which takes a novel approach to curriculum learning. LILAC emphasizes incrementally learning labels instead of incrementally learning difficult samples. It works in two distinct phases: first, in the incremental label introduction phase, we recursively reveal ground-truth labels in small installments while using a fake label for the remaining data. In the adaptive compensation phase, we compensate for failed predictions by adaptively altering the target vector to a smoother distribution. We evaluate LILAC against the closest comparable methods in batch and curriculum learning and label smoothing, across three standard image benchmarks, CIFAR-10, CIFAR-100, and STL-10. We show that our method outperforms batch learning with *higher mean recognition accuracy as well as lower standard deviation in performance consistently across all benchmarks*. We further extend LILAC to show the highest performance on CIFAR-10 for methods using simple data augmentation while exhibiting label-order invariance among other properties.

## 1 INTRODUCTION

Deep networks have seen rich applications in high-dimensional problems characterized by a large number of labels and a high volume of samples. However, successfully training deep networks to solve problems under such conditions is mystifyingly hard (Erhan et al. (2009); Larochelle et al. (2007)). The go-to solution in most cases is Stochastic Gradient Descent with mini-batches (simple batch learning) and its derivatives. While offering a standardized solution, simple batch learning often fails to find solutions that are simultaneously stable, highly generalizable and scalable to large systems (Das et al. (2016); Keskar et al. (2016); Goyal et al. (2017); You et al. (2017)). This is a by-product of how mini-batches are constructed. For example, the uniform prior assumption over datasets emphasizes equal contributions from each data point regardless of the underlying distribution; small batch sizes help achieve more generalizable solutions, but do not scale as well to vast computational resources as large mini-batches. It is hard to construct a solution that is a perfect compromise between all cases.

Two lines of work, curriculum learning and label smoothing, offer alternative strategies to improve learning in deep networks. Curriculum learning, inspired by strategies used for humans (Skinner (1958); Avrahami et al. (1997)), works by gradually increasing the conceptual difficulty of samples used to train deep networks (Bengio et al. (2009); Florensa et al. (2017); Graves et al. (2017)). This has been shown to improve performance on corrupted (Jiang et al. (2017)) and small datasets (Fan et al. (2018)). More recently, deep networks have been used to categorize samples (Weinshall et al. (2018)) and variations on the pace with which these samples were shown to deep networks were analyzed in-depth (Hacohen & Weinshall (2019)). To the best of our knowledge, previous works assumed that samples cover a broad spectrum of difficulty and hence need to be categorized and presented in a specific order. This introduces computational overheads e.g. pre-computing the relative difficulty of samples, and also reduces the effective amount of data from which a model can

learn in early epochs. Further, curriculum learning approaches have not been shown to compete with simple training strategies at the top end of performance in image benchmarks.

A complementary approach to obtaining generalizable solutions is to avoid over-fitting or getting stuck in local minima. In this regard, label smoothing offers an important solution that is invariant to the underlying architecture. Early works like Xie et al. (2016) replace ground-truth labels with noise while Reed et al. (2014) uses other models' outputs to prevent over-fitting. This idea was extended in Bagherinezhad et al. (2018) to an iterative method which uses logits obtained from previously trained versions of the same deep network. While Miyato et al. (2015) use local distributional smoothness, based on the robustness of a model's distribution around a data point, to regularize outcomes, Pereyra et al. (2017) penalized highly confident outputs directly. Closest in spirit to our work is the label smoothing method defined in Szegedy et al. (2016), which offers an alternative target distribution for all training samples with no extra data augmentation. In general, label smoothing is applied to all examples regardless of how it affects the network's understanding of them. Further, in methods which use other models to provide logits/labels, often the parent network used to provide those labels is trained using an alternate objective function or needs to be fully re-trained on the current dataset, both of which introduce additional computation.

In this work, we propose LILAC, *Learning with Incremental Labels and Adaptive Compensation*, which emphasizes a label-based curriculum and adaptive compensation, to improve upon previous methods and obtain *highly accurate* and *stable* solutions. LILAC is conceived as a method to learn strong embeddings by using the recursive training strategy of incremental learning alongside the use of unlabelled/wrongly-labelled data as hard negative examples. It works in two key phases, 1) **incremental label introduction** and 2) **adaptive compensation**.

In the first phase, we incrementally introduce groups of labels in the training process. Data, corresponding to labels not yet introduced to the model, use a single fake label selected from within the dataset. Once a network has been trained for a fixed number of epochs with this setup, an additional set of ground-truth labels is introduced to the network and the training process continues. In recursively revealing labels, LILAC allows the model sufficient time to develop a strong understanding of each class by contrasting against a large and diverse set of negative examples.

Once all ground-truth labels are revealed the adaptive compensation phase of training is initiated. This phase mirrors conventional batch learning, except we adaptively replace the target one-hot vector of incorrectly classified samples with a softer distribution. Thus, we avoid adjusting labels across the entire dataset, like previous methods, while elevating the stability and average performance of the model. Further, instead of being pre-computed by an alternative model, these softer distributions are generated on-the-fly from the outputs of the model being trained. We apply LILAC to three standard image benchmarks and compare its performance to the strongest known baselines.

While incremental and continual learning work on evolving data distributions with the addition of memory constraints ((Rebuffi et al., 2017; Castro et al., 2018) and derivative works), knowledge distillation ((Schwarz et al., 2018; Rolnick et al., 2018) and similar works) or other requirements, this work is a departure into using negative mining and focused training to improve learning on a fully available dataset. In incremental/continual learning works, often the amount of data used to retrain the network is small compared to the original dataset while in LILAC we fully use the entire dataset, distinguished by Seen and Unseen labels. Thus, it avoids data deficient learning. Further, works like Bucher et al. (2016); Li et al. (2013); Wang & Gupta (2015) emphasize the importance of hard negative mining, both in size and diversity, in improving learning. Although the original formulation of negative mining was based on imbalanced data, recent object detection works have highlighted its importance in contrasting and improving learning in neural networks.

To summarize, our main contributions in LILAC are as follows,

- we introduce a new take on curriculum learning by incrementally learning *labels* as opposed to *samples*,
- our method adaptively compensates incorrectly labelled samples by softening their target distribution which *improves performance* and *removes external computational overheads*,
- we improve average recognition accuracy and decrease the standard deviation of performance across several image classification benchmarks compared to batch learning, a property not shared by other curriculum learning and label smoothing methods.

## 2 LILAC

In LILAC, our main focus is to induce better learning in deep networks. Instead of the conventional curriculum learning approach of ranking samples, we consider all samples equally beneficial. Early on, we focus on learning labels in fixed increments (Section 2.1). Once the network has had a chance to learn all the labels, we shift to regularizing the network to prevent over-fitting by providing a softer distribution as the target vector for previously misclassified samples (Section 2.2). An overview of the entire algorithm discussed is available in the appendix as Algorithm 1.

### 2.1 INCREMENTAL LABEL INTRODUCTION PHASE

In the incremental phase, we initially replace the ground-truth labels of several class using a constant held-out label. Gradually, over the course of several fixed intervals of training we reveal the true label. Within a fixed interval of training, we keep constant two sets of data, "Seen", whose ground-truth labels are known and "Unseen", whose labels are replaced by a fake value. When training,

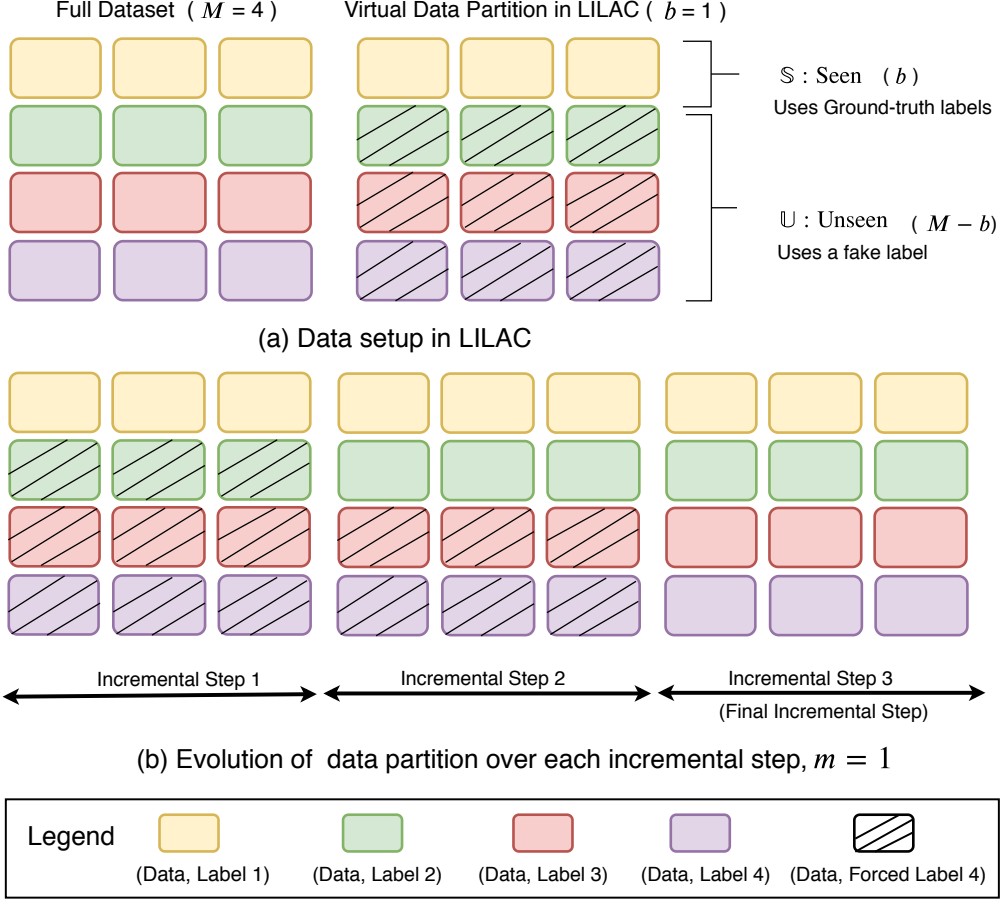

(a) Data setup in LILAC

(b) Evolution of data partition over each incremental step, $m = 1$

Figure 1: (**Top**) In LILAC, data is virtually partitioned into either $\mathbb{S}$ : Seen or $\mathbb{U}$ : Unseen. Data under $\mathbb{S}$ use their ground-truth labels while data in $\mathbb{U}$ use a designated fixed unseen label. (**Bottom**) Illustration of the evolution of data partitions in the incremental label introduction phase for a four label dataset. In the first incremental step, only one label is used for training while the remaining data use label 4. A short period of training is performed with this fixed setup, where data from $\mathbb{U}$ is uniformly sampled to match the number of samples from $\mathbb{S}$, in every mini-batch. The final incremental step depicted is equivalent to batch learning since all the labels are available to the network. Once all the ground-truth labels are revealed we begin the adaptive compensation phase described in Sec. 2.2.

mini-batches are uniformly sampled from the entire training set, but the instances from "Unseen" classes use the held-out label. By the end of the final interval, we reveal all ground-truth labels.

We now describe the incremental phase in more detail. At the beginning of the incremental label introduction phase, we virtually partition data into two mutually exclusive sets, $\mathbb{S}$ : Seen and $\mathbb{U}$ : Unseen, as shown in Fig. 1. Data samples in $\mathbb{S}$ use their ground-truth labels as target values while those in $\mathbb{U}$ use a designated unseen label, which is held constant throughout the entire training process. LILAC assumes a random ordering of labels, $Or(M)$, where $M$ denotes the total number of labels in the dataset. Within this ordering, the number of labels and corresponding data initially placed in $\mathbb{S}$ is defined by the variable $b$. The remaining labels, $M - b$, are initially placed in $\mathbb{U}$ and incrementally revealed in intervals of $m$ labels, a hyper-parameter defined by the user.

Training in the incremental phase happens at fixed intervals of $E$ epochs each. Within a fixed interval, the virtual data partition is held constant. Every mini-batch of data is sampled uniformly from the entire original dataset and within each mini-batch, labels are obtained based on their placement in $\mathbb{S}$ or $\mathbb{U}$. Then the number of samples from $\mathbb{U}$ is reduced or augmented, using a uniform prior, to match the number of samples from $\mathbb{S}$. This is done to ensure no unfair skew in predictions towards $\mathbb{U}$ since all data points use the same designated label. Finally, the curated mini-batches of data are used to train the neural network. At the end of each fixed interval, we reveal another set of $m$ ground-truth labels and move samples of those classes from $\mathbb{U}$ to $\mathbb{S}$ after which the entire data curation and training process is repeated for the next interval.

## 2.2 ADAPTIVE COMPENSATION

Once all the ground-truth labels are available to the deep network, we begin the adaptive compensation phase of training. The main idea behind adaptive compensation is, if the network is unable to correctly predict a sample's label even after allowing sufficient training time, then we alter the target vector to a less peaked distribution. Compared to learning one-hot vectors, this softer distribution can more easily be learned by the network. Unlike prior methods we adaptively modify the target vector only for incorrectly classified samples on-the-fly.

In this phase, the network is trained for a small number of epochs using standard batch learning. Let $T$ be the total number of training epochs in the incremental phase and batch learning. During the adaptive compensation phase, we start at epoch $e$, where $e > T$. For a mini-batch of samples in epoch $e$, predictions from the model at $e - 1$ are used to determine the final target vector used in the objective function; specifically, we soften the target vector for an instance iff it was misclassified by the model at the end of epoch $e - 1$. The final target vector for the $i$th instance at epoch $e$, $t_{e,i}$, is computed based on the model $\phi_{e-1}$ using Equation 1.

$$
t_{e,i} = \begin{cases} (\frac{\epsilon M - 1}{M-1})\delta_{y_i} + (\frac{1-\epsilon}{M-1})\mathbb{1}, & \arg\max(\phi_{e-1}(x_i)) \neq y^i \\ \delta_{y_i}, & \text{otherwise} \end{cases} . \tag{1}
$$

Here, $(x_i, y_i)$ denote a training sample and its corresponding ground-truth label for sample index $i$ while $\delta_{y_i}$ represents the corresponding one-hot vector. $\mathbb{1}$ is a vector of $M$ dimensions with all entries as 1 and $\epsilon$ is a scaling hyper-parameter.

## 3 EXPERIMENTS

**Datasets** We use three datasets, CIFAR-10, CIFAR-100 (Krizhevsky et al. (2009)) and STL-10 (Coates et al. (2011)), to evaluate our method and validate our claims. CIFAR-10 and CIFAR-100 are 10 and 100 class variants of the popular image benchmark CIFAR. Each of these contains 50,000 images in the training set and 10,000 images in the testing set. STL-10 is a 10 class subset of ImageNet with 500 and 800 samples per class for training and testing subsets, respectively.

**Metrics** The common metric used to evaluate the performance of all the learning algorithms is *average recognition accuracy(%)* and *standard deviation* across 5 trials. We also report *consistency*, which is a binary metric that indicates whether the training strategy results in higher average performance and lower standard deviation compared to standard batch learning across all datasets.

Table 1: Performance comparison across all comparable methods and datasets. Under similar setups, without any extra overheads in computation, LILAC consistently achieves higher accuracy and lower std. dev. than batch learning across all compared datasets, which is not the case for any other baseline. The highlighted performances represent the best case scenario which indicate higher average performance combined with lower standard deviation compared to batch learning.

| Training | Performance (%) | | | Consistency |
| | CIFAR 10 | CIFAR 100 | STL 10 | |
|---|---|---|---|---|
| Batch Learning | $95.28 \pm 0.141$ | $78.51 \pm 0.145$ | $73.68 \pm 0.764$ | N/A |
| Fixed Curriculum | $\mathbf{95.32 \pm 0.099}$ | $77.36 \pm 0.412$ | $72.13 \pm 0.953$ | ✗ |
| Label Smoothing | $\mathbf{95.39 \pm 0.050}$ | $79.05 \pm 0.216$ | $72.26 \pm 0.638$ | ✗ |
| Dynamic Batch Size (DBS) | $\mathbf{95.32 \pm 0.022}$ | $78.98 \pm 0.206$ | $73.20 \pm 0.656$ | ✗ |
| Random Augmentation (RA) | $95.22 \pm 0.060$ | $72.49 \pm 0.467$ | $73.63 \pm 0.387$ | ✗ |
| LILAC w/o AC (**ours**) | $95.26 \pm 0.110$ | $78.44 \pm 0.227$ | $73.53 \pm 0.331$ | ✗ |
| LS + LILAC (**ours**) | $\mathbf{95.39 \pm 0.088}$ | $79.07 \pm 0.171$ | $\mathbf{73.80 \pm 0.714}$ | ✗ |
| LILAC (**ours**) | $\mathbf{95.35 \pm 0.098}$ | $\mathbf{78.73 \pm 0.127}$ | $\mathbf{74.35 \pm 0.568}$ | ✓ |

**Experimental Setup**  For CIFAR-10/100, we use ResNet18 (He et al. (2016)) as the architectural backbone for all methods; for STL-10, we use ResNet34. In each interval of LILAC's incremental phase, we train the model for 10 epochs each for CIFAR-10/100, and 5 epochs each for STL-10. During these incremental steps, we use a learning rate of 0.1, 0.01 and 0.1 for CIFAR-10, CIFAR-100, and STL-10 respectively. The standard batch learning settings used across all datasets are listed in the appendix. These settings reflect the setup used in LILAC once the incremental portion of training is complete and the algorithm moves into the adaptive compensation phase. Within this phase epochs 175, 525 and 120 are used as thresholds (epoch $T$) for CIFAR-10, 100 and STL-10 respectively.

**Baselines**

- Stochastic gradient descent with mini-batches is the baseline against which all methods are compared.

- Curriculum learning (Bengio et al., 2009) forms a family of related works which aim to help models learn *faster* and optimize to a *better minimum*. Following the methodology proposed in this work we artificially create a subset of the dataset called "Simple", by selecting data that is within a value of 1.1 as predicted by a linear one-vs-all SVR model trained to regress to the ground-truth label. The deep network is trained on the "Simple" dataset for a fixed period of time that mirrors the total number of epochs of the incremental phase of LILAC after which the entire dataset is used to train the network.

- Label Smoothing (Szegedy et al., 2016) is the closest relevant work to use label smoothing as a form of regularization without extra data augmentation. This non-invasive baseline is used as a measure of the importance of regularization and for its ability to boost performance.

- Dynamic Batch Size (DBS) is a custom baseline used to highlight the importance of variable batch size in training neural networks. DBS randomly copies data available within a mini-batch to mimic variable batch size. Further, all ground-truth labels are available to this model throughout the training process.

- Random Augmentation (RA) is a custom baseline used to highlight the importance of virtual data partitions in LILAC. Its implementation closely follows LILAC but excludes the adaptive compensation phase. The main difference between LILAC and RA is that RA uses data from a one randomly chosen class, in $\mathbb{U}$, within a mini-batch while data from all classes in $\mathbb{U}$ is used in LILAC to equalize the number of samples from $\mathbb{S}$ and $\mathbb{U}$.

## 3.1 STANDARDIZED COMPARISON RESULTS

Table 1 clearly illustrates *improvements in average recognition accuracy, decrease in standard deviation* and *consistency* when compared to batch learning. While certain setups have the highest

Table 2: Comparison of LILAC's performance against top performing algorithms on CIFAR-10 with *standard pre-processing (random crop + flip)*. Our method easily outperforms both the base shake-drop network (Yamada et al. (2018)) as well as previous methods.

| Method | CIFAR-10 |
|---|---|
| Wide Residual Networks (Zagoruyko & Komodakis, 2016) | 96.11 |
| Multilevel Residual Networks (Zhang et al., 2017) | 96.23 |
| Fractional Max-pooling (Graham, 2014) | 96.53 |
| Densely Connected Convolutional Networks (Huang et al., 2017) | 96.54 |
| Drop-Activation (Liang et al., 2018) | 96.55 |
| Shake-Drop (Yamada et al., 2018) | 96.59 |
| **Shake-Drop + LILAC (ours)** | **96.79** |

Table 3: Impact of different phases on the final performance of LILAC. Rows 1 and 2 compare batch learning to LILAC without adaptive compensation while rows 3 and 4 highlight the impact of adding adaptive compensation. While incremental label introduction on its own is not a stand-out performer, adding adaptive compensation to it improves performance beyond standard batch learning. The highlighted values indicate higher average performance combined with lower standard deviation compared to batch learning.

| Training | Performance (%) | | |
| | CIFAR 10 | CIFAR 100 | STL 10 |
|---|---|---|---|
| Batch | $95.28 \pm 0.141$ | $78.51 \pm 0.145$ | $73.68 \pm 0.764$ |
| LILAC w/o AC (**ours**) | $95.26 \pm 0.110$ | $78.44 \pm 0.227$ | $73.53 \pm 0.331$ |
| Batch + AC | $\mathbf{95.38 \pm 0.136}$ | $78.54 \pm 0.175$ | $73.32 \pm 0.581$ |
| LILAC (**ours**) | $\mathbf{95.35 \pm 0.098}$ | $\mathbf{78.73 \pm 0.127}$ | $\mathbf{74.35 \pm 0.568}$ |

performance on specific datasets (e.g., Label Smoothing on CIFAR-10/100), they are not consistent across all datasets and do not find more stable solutions than LILAC (std. of 0.216 compared to 0.127 from LILAC) LILAC is able to achieve superior performance without unnecessary overheads such as computing sample difficulty or irreversibly altering the ground-truth distribution across all samples.

A key takeaway from DBS is the relative drop in standard deviation combined with higher average performance when compared to baselines like fixed curriculum and label smoothing. RA serves to highlight the importance of harvesting data from all classes in $\mathbb{U}$ simultaneously, for "negative" samples. The variety of data to learn from provides a boost in performance and standard deviation across the board in LILAC w/o AC as opposed to RA. DBS and RA underline the importance of variable batch size and data partitioning in the makeup of LILAC.

We further extend LILAC to train the base pyramidnet with shake-drop regularization ($p = 1.0$) (Yamada et al. (2018)). From Table 2 we clearly see that LILAC can be extended to provide the highest performance on CIFAR-10 given a standard preprocessing setup. To provide a fair comparison we highlight top performing methods with standard preprocessing setups that avoid partial inputs (at the node or image level) since LILAC was developed with fully available inputs in mind. *Across all these learning schemes, LILAC is the only one to consistently increase classification accuracy and decrease the standard deviation across all datasets compared to batch learning.*

## 3.2 ABLATION: BREAKDOWN OF LILAC'S PHASES

Fig. 2 illustrates the evolution of the embedding across the span of the incremental phase. This space has more degrees of separation when compared to an equivalent epoch of training with batch learning where all the labels are available. Table 3 provides a breakdown of the contribution of each phase of LILAC and how they combine to elevate the final performance. Here, in LILAC w/o AC we replace the entire AC phase with simple batch learning while in Batch + AC we include adaptive compensation with adjusted thresholds. The first half of this table compares the impact of incre-

CIFAR-10

CIFAR-100

STL-10

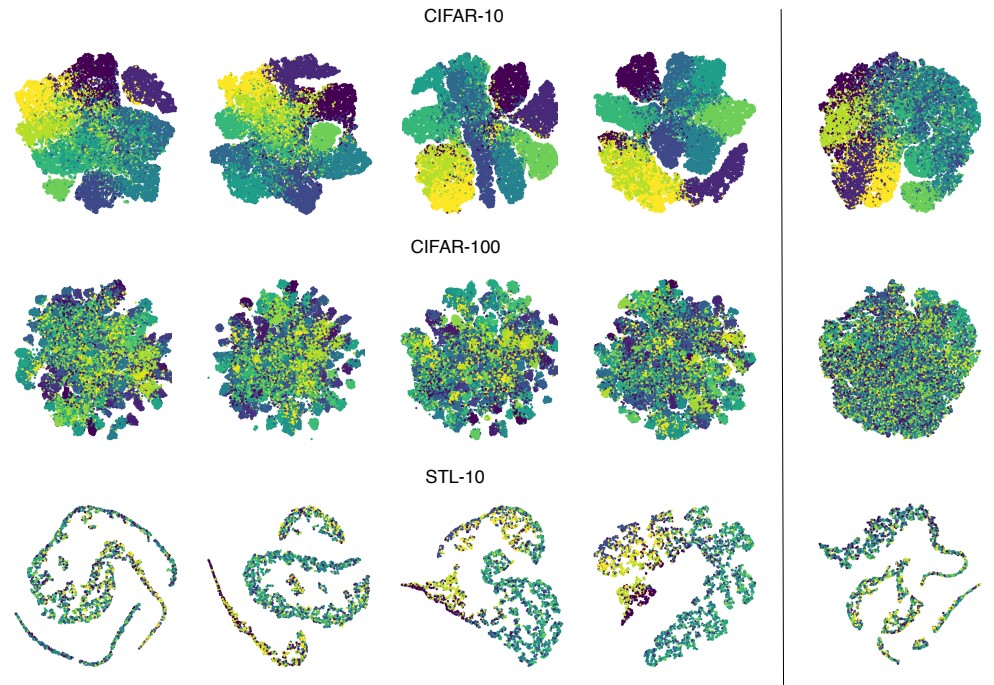

(a) Evolution of representation space after 1/4, 1/2, 3/4 and full span of incremental phase      (b) An epoch of batch learning

Figure 2: Side-by-side comparison between the representation spaces learned by LILAC and batch learning. Through the entire incremental label introduction phase, the representation space evolves to being more well spaced. Images in column 4 and 5 show comparable points in training space when all labels are available to the deep network being trained. These images support our claim that the deep network starts at a better initialization than standard batch learning, whose effect is carried throughout the training process.

mentally introducing labels to a deep network against standard batch learning. We clearly observe that performances across Rows 1 and 2 fall within the indicated standard deviation of each other. However, from Fig. 2 we know that LILAC start from a qualitatively better solution. Combining these results, we conclude that the emphasis on a lengthy secondary batch learning phase erodes overall performance.

The second half of Table 3 shows the impact of adding adaptive compensation on batch learning and LILAC. When added to standard batch learning there isn't a clear and conclusive indicator of improvement across all benchmarks in both average performance and standard deviation. However, in combination with the incremental label introduction phase of LILAC, adaptive compensation improves average performance as well as decreases standard deviation, indicating an improved stability and consistency. Given the similarity in learning between the batch setup and LILAC, when all labels have been introduced, we show that the embedding space learned by incrementally introducing labels (Fig. 2) is distinct from standard batch learning and is more amenable to AC.

### 3.3 PROPERTIES OF LILAC

Through previous experiments we have established the general applicability of LILAC while contrasting its contributions to that of standard batch learning. In this section we dive deeper to reveal some characteristics of LILAC that further supplement the claim of general applicability. Specifically, we characterize the impact of label ordering, smoothness of alternate target vector distribution and injection of larger groups of labels in the incremental phase.

**Ordering of Labels** Throughout the standard experiments, we assume labels are used in the ascending order of value. When this is modified to a random ordering or in ascending order of diffi-

Table 4: (**Top**) This table illustrates the impact of using random label order vs. ascending vs. ascending difficulty on LILAC w/o AC. LILAC doesn't provide any explicit pattern in the outcomes. (**Middle**) Varying the smoothness of the alternate target vector causes slight variations in performance. (**Bottom**) Introducing multiple labels per incremental learning interval hurts performance. The learned embedding space is different, which leads to diverging performances. The introduction of a lower number of label allows for thorough learning and improved final performance in LILAC. Highlighted numbers are peak average performance values.

| Property | Training | Performance (%) | | |
| --- | --- | --- | --- | --- |
| | | CIFAR-10 | CIFAR-100 | STL-10 |
| Label Order: Rnd. | LILAC w/o AC | $95.26 \pm 0.065$ | $78.27 \pm 0.122$ | $73.74 \pm 0.846$ |
| Label Order: Asc. | | $95.26 \pm 0.110$ | $78.44 \pm 0.227$ | $73.53 \pm 0.331$ |
| Label Order: Asc. Difficulty | | $95.42 \pm 0.089$ | $78.23 \pm 0.084$ | $73.31 \pm 1.117$ |
| $\epsilon = 1.0$ | | $95.25 \pm 0.111$ | $78.33 \pm 0.178$ | $73.68 \pm 0.815$ |
| $\epsilon = 0.9$ | | $95.31 \pm 0.073$ | $78.46 \pm 0.211$ | $73.26 \pm 1.285$ |
| $\epsilon = 0.8$ | | $95.25 \pm 0.088$ | $78.58 \pm 0.364$ | $73.47 \pm 0.734$ |
| $\epsilon = 0.7$ | LILAC | $95.25 \pm 0.129$ | $78.71 \pm 0.256$ | $73.27 \pm 0.179$ |
| $\epsilon = 0.6$ | | $95.33 \pm 0.073$ | $78.38 \pm 0.257$ | $73.71 \pm 0.421$ |
| $\epsilon = 0.5$ | | $\mathbf{95.35 \pm 0.098}$ | $\mathbf{78.73 \pm 0.127}$ | $\mathbf{74.35 \pm 0.568}$ |
| $\epsilon = 0.4$ | | $95.30 \pm 0.127$ | $78.62 \pm 0.208$ | $73.38 \pm 1.296$ |
| Label Group ($m$): 1 | | $\mathbf{95.35 \pm 0.098}$ | $\mathbf{78.73 \pm 0.127}$ | $\mathbf{74.35 \pm 0.568}$ |
| Label Group ($m$): 2 (5) | LILAC | $95.22 \pm 0.175$ | $78.73 \pm 0.215$ | $72.95 \pm 0.372$ |
| Label Group ($m$): 3 (10) | | $95.29 \pm 0.189$ | $78.47 \pm 0.275$ | $73.61 \pm 0.771$ |

culty, results from Table 4 suggest that there is no explicit benefit or pattern. Other than the extra impact of continually fluctuating label orders across trials, there isn't a large gap in performance. Thus, we claim LILAC is relatively invariant to the order of label introduction.

**Smoothness of Target Vector in Adaptive Compensation**     During adaptive compensation, $\epsilon = 0.5$ is used in the alternate target vector for samples with failed predictions throughout all experiments in Sections 3.1 and 3.2. When extended to a variety of $\epsilon$ values, we observe that most variations of the peak performance still fall within the standard deviation range for each dataset. However, the peak average performance values usually occur between 0.7 to 0.5.

**Injection of Label Groups**     While LILAC was designed to allow the introduction of multiple labels in a single incremental step, throughout the experiments in Sections 3.1 and 3.2 only 1 label was introduced per step to allow thorough learning while eliminating the chance of conflicting decision boundaries from multiple labels. Revealing multiple labels instead of 1 label per incremental step has a negative impact on the overall performance of the model. Table 4 shows that adding large groups of labels force lower performance, which is in line with our hypothesis that revealing fewer labels per incremental step makes the embedding space more amenable to adaptive compensation.

## 4    CONCLUSION

In this work, we proposed LILAC which rethinks curriculum learning based on incrementally learning labels instead of samples. This approach helps kick-start the learning process from a substantially better starting point while making the learned embedding space amenable to adaptive negative logit compensation. Both these techniques combine well in LILAC to show the highest performance on CIFAR-10 for simple data augmentations while easily outperforming batch and curriculum learning and label smoothing on comparable network architectures. The next step in unlocking the full potential of this setup is to extend this setup to include a confidence measure on the predictions of network so that it can handle the effects of dropout or partial inputs. In further expanding LILAC's ability to handle partial inputs, we aim to explore its effect on standard incremental learning (memory constrained) while also extending it applicability to more complex neural network architectures.

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

## A    LILAC: ALGORITHM

---

**Algorithm 1:** Training strategy inspired by incremental learning

---
initialization;
**Input:** $(X, Y)$ **where** $Y \in c_1, c_2, .., c_M$;
$M =$ **Total number of labels in the dataset;**
$m =$ **Number of labels to introduce in 1 incremental step;**
$n =$ **Total number of samples;**
$b =$ **Starting incremental batch;**
$C = \{c_1, c_2, .., c_M\}$;
**for** *inc_batch= b to* $\left(\frac{M}{m}\right)$ **do**
    **for** *fixed epochs e* **do**
        $\widetilde{C} = \{c_1, .., c_{(inc\_batch \times \frac{M}{m} + m)}\}$;
        $n_{\widetilde{C}} =$ **number of samples with labels in** $\widetilde{C}$;
        $\mathbb{S} = \{(x_i, y_i) | y_i \in \widetilde{C}\}_{i=1}^{n_{\widetilde{C}}}$;
        $\mathbb{U} = \{(x_j, y_j) | y_j \in C/\widetilde{C}\}_{j=1}^{n - n_{\widetilde{C}}}$;
        $F' \subseteq \mathbb{U} : F' = \{(x_j, y_j) | y_j \in C/\widetilde{C}\}_{j=1}^{n_{\widetilde{C}}}$ **selected at random;**
        **s.t.** $|\mathbb{S}| = |F'|$ ;
        **if** *inc_batch* $= \frac{M}{m}$ *and* $e \geq \delta$ **then**
            $F' =$ **update target vectors using Eqn. 1** ;
        **end**
        **Train Model with data** $(\mathbb{S} \cup F')$
    **end**
**end**

---

## B    HYPER-PARAMETER SETUPS

Table 5: List of hyper-parameters used to in batch learning. Note: All experiments used the SGD optimizer.

| Parameters | CIFAR10/100 | STL10 |
|---|---|---|
| Epochs | 300 | 450 |
| Batch Size | 128 | 128 |
| Learning Rate | 0.1 | 0.1 |
| Lr Milestones | [90 180 260] | [300 400] |
| Weight Decay | 0.0005 | 0.0005 |
| Nesterov Momentum | Yes | Yes |
| Gamma | 0.2 | 0.1 |

Table 6: List of hyper-parameters used in tSNE.

| Parameters | Value |
|---|---|
| # Components | 2 |
| Perplexity | 30.0 |
| Early Exaggeration | 12.0 |
| Lr | 200.0 |
| Iterations | 1000 |
| Iterations w/o Progress | 300 |
| Min. Grad. Norm | 1e-07 |
| Metric | Euclidean |
| Method | Barnes-Hut |
| Angle | 0.5 |

## C  APPLICABILITY TO VIDEOS

Table 7: We extend LILAC to a 3D convolutional architecture and the HMDB51 (video) dataset. We clearly observe an improvement in over performance when compared to batch learning.

| Model | Training | HMDB51 Performance (%) |
|---|---|---|
| C3D | Batch | 45.52 |
| | LILAC w/o N_L | 45.29 |
| C3D | Batch + N_L | 46.14 |
| | LILAC | **46.17** |

## D  PROPERTY: VARIATION OF FIXED INTERVAL SIZE IN INCREMENTAL LABEL INTRODUCTION

Table 8: The table captures the effect of varying the number of epochs used for the fixed training intervals in the incremental label introduction phase. Across CIFAR-10 there is an obvious peak after which the mean value decreases. However, in STL-10 there seems to be a consistent increase, with the assumption of minor noise. Finally, in CIFAR-100 there isn't a clear pattern.

| Property | Training | Performance (%) | | |
|---|---|---|---|---|
| | | CIFAR-10 | CIFAR-100 | STL-10 |
| E = 1 | | $95.20 \pm 0.042$ | $78.39 \pm 0.221$ | $73.02 \pm 0.949$ |
| E = 3 | | $95.19 \pm 0.138$ | $78.59 \pm 0.147$ | $73.26 \pm 1.180$ |
| E = 5 | LILAC w/o AC | $95.33 \pm 0.195$ | $78.30 \pm 0.274$ | $73.53 \pm 0.331$ |
| E = 7 | | $95.37 \pm 0.167$ | $78.51 \pm 0.209$ | $73.23 \pm 0.864$ |
| E = 10 | | $95.26 \pm 0.110$ | $78.44 \pm 0.227$ | $73.89 \pm 0.603$ |

From the results in Table 8, we observe that the choice of E is dependent on the dataset. There isn't an explicit pattern that can be used to select the value of E without trial runs. Further, the available run-time is an important constraint when select E from a range of values since both $m$ and E affect it.

