# OpenReview forum: "Rethinking Curriculum Learning With Incremental Labels And Adaptive Compensation"
_ICLR.cc/2020/Conference — Reject_

### Official Review · AnonReviewer1 · 2019-10-21
**Official Blind Review #1**

**Rating:** 3

**Review:**

1. Summary:

This paper proposes a novel direction for curriculum learning.  Previous works in the area of curriculum learning focused on choosing easier samples first and harder samples later when learning the neural network models.  This is problematic since we need to first compute how difficult each samples are, which introduces computational overheads.  In this work, the paper propose to gradually learn with a class-wise perspective instead.  The neural network has only access to the labels of certain classes (chosen randomly) in the beginning, and the samples that belong to the rest of the classes are treated as unseen samples but with a label forced into the last class.  Then, the true labels of unseen classes are gradually revealed, and this is repeated until in the final incremental step, all labels are revealed.  The method further has an adaptive compensation step, which use a less peaked distribution label for supervision only for the incorrectly predicted samples.  The experiments show that with only the first step, the proposed method is worse than the original batch learning, but by adding the second label smoothing step, there is improvement over the original batch learning setup.


2. Decision:

Weak Reject -- The class-wise idea for curriculum learning is interesting but the motivation and intuition behind the design of the proposed method is weak.


3. Reasons for decision:

Pros:

The class-wise idea used in this paper seems to add a new direction to the area of curriculum learning.  The experiments are well designed, with ablation study to understand the behavior of the proposed method in depth.

Cons:

The motivation behind the ideas of the paper and the design of the procedure was not so clear.  Why would it be beneficial to start learning from a few classes in the beginning and then gradually expanding the class labels?  I was also curious if this will be more beneficial with a dataset with more classes (maybe not so different by looking at the comparison of CIFAR10 and 100 according to Table 1).

If my understanding was correct, the masked labels from unseen samples are all forced into the last class.  Does this design cause new issues, since the last class will have a lot of label noise until the final incremental step?  If yes, is it possible to consider other ways to add a fake label?   A naive way might be: for unseen samples, attach a fake soft label that has uniform probability over masked classes (M-b) and zero probability for the seen classes (in b).

In the experiments, it seems that the proposed method has consistency but no other baselines have consistency.  However, if you just look at the best mean accuracy, a naive label smoothing is often better than the proposed method.

4. Additional comments not related to the decision of the paper:

In Algorithm 1 in appendix A, the equation link is lost: “Eqn. ??”

In appendix B, what is the decay rate for LR milestones?



After response:

Thank you for answering my questions and for the comments.  Even though towards the end of the learning stage, the true labels are given for all samples, I feel these noisy steps go against the spirit of curriculum learning.  Starting from small classes seems to be a good curriculum and is intuitive, but the noisy part seems to be adding a bad curriculum.  This paper can be much stronger if this part is solved.

**Experience Assessment:**

I have read many papers in this area.

**Review Assessment: Checking Correctness Of Derivations And Theory:**

N/A

**Review Assessment: Checking Correctness Of Experiments:**

I assessed the sensibility of the experiments.

**Review Assessment: Thoroughness In Paper Reading:**

I read the paper at least twice and used my best judgement in assessing the paper.

---

> ### Author Response · Authors · 2019-11-11
> **Description of Motivation and Response to Reviewer's comments**
>
> We thank the reviewer for their detailed feedback on LILAC. Below, we paraphrase the reviewer's comments and address them in full.
> 1. Clarify the motivation/intuition behind LILAC
> Ans. LILAC was conceived as a method to learn strong embeddings by using the recursive training process from incremental learning alongside the use of unlabelled/wrongly-labelled data as contrastive negative examples.
> At its core, LILAC tries to learn stronger embeddings by pitting a small number of classes against the entire remaining dataset for a fixed learning interval.
> In recursively unmasking labels, the model has sufficient time to develop a strong understanding of each class while gently introducing a small number of new labels to the model at every stage.
> At the end of the incremental label introduction phase, on comparing the starting points when the model has knowledge of the entire dataset, we observe that optimization starts from a more well-structured point in the solution space in LILAC than in the case of random initialization (Figure 2).
> Secondarily, LILAC takes advantage of its learned embeddings by using it to identify misclassified samples and softening their target vectors to reduce the number of misclassifications by gently adapt the embedding space to incorporate them. Softening target vectors tries to make learning simpler.
>
> 2. Why is it beneficial to start learning from a small number of classes and expand gradually? Is it useful for datasets with more labels?
> Ans. The main benefit of starting learning from a few classes and then gradually expanding is three-fold.
> a. The amount of time spent in learning about data from a small subset of classes and contrasting it against an Unseen counter-part helps develop a stronger understanding.
> b. By keeping a sufficiently large portion of the dataset in the unknown/weakly-supervised set, the diversity of data being compared against helps the model learn stronger features initially.
> c. Further, it reduces the interference of multiple labels that would otherwise exist. Instead, the model can focus on the seen label set while reducing the entirety of the remaining data into 1 common label.
> Across the 4 datasets on which we test our method, LILAC is beneficial to all of them. It is most important when the performance has not been saturated at the top end.
>
> 3. Does forcing all masked labels into 1 value bring up issues? If so, can you add a fake soft label with a uniform prior over unseen classes and 0 probability over seen classes as an alternative?
> Ans. Forcing all masked labels into 1 value, regardless of the value itself, doesn't bring up many issues. This is primarily because, during the incremental label introduction phase, we recursively unmask ground-truth labels. This ensures that the true (data, label) pairing is eventually available to the model during training, including an extended period of time in the AC phase.
> The suggestion of using a fake soft label with a uniform prior over unseen classes is an interesting one.
> a. We believe that the use of a flexible soft label would force a large swing in gradients, repeatedly, especially since the size of the Unseen set is large when compared to the size of data pertaining to each seen label.
> b. We used a fixed soft label so that the model can relegate learning its explicit identity and instead focus on using it to contrast against data from the Seen subset. By providing a uniform prior on the label of the unseen set, we could possibly force to the model learn the explicit identity of the unseen set repeatedly and this could negatively impact the model.
>
> 4. The proposed method has consistency. However, naive label smoothing has better mean accuracy (often) than the proposed method.
> Ans. The proposed method’s performance mentioned in all the tables is built on the assumption that a one-hot vector is used for the ground-truth target distribution. If we apply simple label smoothing’s alternate vector as the ground-truth vector and apply LILAC, we are able to achieve and in certain cases surpass the peak mean accuracy achieved by simple label smoothing while retaining a smaller standard deviation (in CIFAR-100 and STL-10).
>
> Method			                   CIFAR-10          CIFAR-100            STL-10
> Label Smoothing: 		95.39 ± 0.050    79.05 ± 0.216    72.26 ± 0.638
> LILAC + Label smoothing:  95.39 ± 0.088    79.07 ± 0.171    73.65 ± 0.607
>
> 5. Appendix B, what is the decay rate for LR milestones?
> Ans. The same weight decay rate carries through to the LR milestones. As for the decrease in the learning rate, gamma is the multiplicative factor by which the learning rate is decreased.

---

### Official Review · AnonReviewer2 · 2019-10-24
**Official Blind Review #2**

**Rating:** 1

**Review:**

---- Paper summary ----
This paper proposes a curriculum learning approach for classification. The proposed curriculum consists of two phases:
(1) a “label introduction phase”, in which the model is able to see and learn to classify only subset of labels (the model still trains on the samples belonging to the “unseen” classes, but their label is now set to a default class). The subset of seen labels is expanded incrementally, until the entire label set is observed.
(2) an “adaptive compensation phase”, where the model trains on all labels, but the targets for each class are replaced from 1-hot vectors to smooth version. This only applies to the classes on which the model has made mistakes in a previous training round.
This method is tested on 3 image classification datasets, and a single neural network architecture is tested per dataset (either ResNet18 or ResNet34).

---- Overall opinion ----
While the ideas introduced in this paper may have merit, I believe the experimental evidence is quite limited. Based on the results shown in the paper, I am not convinced that this approach is better than the baselines it compares to. Moreover, since the claim is that this approach is a general curriculum learning method, I find the setting it was tested on very limited (3 datasets, 1 model per dataset), especially since there are no theoretical results to complement the empirical evidence. Finally, the method introduces several parameters (b, m, E, T, eta) that are treated is a somewhat hand-wavy manner, without a proper analysis on the effect of such parameters and how one should set them. Details on this, and other major issues, can be found below. For these reasons, I believe the paper in its current form is not yet ready for publication.

---- Major issues ----
1. The paper simply mentions that the unseen labels are set to a default label, which Figure 1 implies (and is not otherwise clarified) is one of the labels in the dataset. I am not sure intuitively why it makes sense to force the model in the beginning to map a sample’s inputs to another label from the dataset, which in the end it has to learn is wrong. Doesn’t it make more sense to map the unseen labels to a new, fictional label? If not, then how do you decide which of the M labels to choose as the fake label?

2. The method introduces several hyperparameters, such as b (number of visible labels in the beginning), m (number of labels to reveal at each step), E (number of epochs in each incremental phase), T, epsilon. The specific numbers used in the experiments are reported, but it is not clear how these were chosen, and how one would choose them for a new dataset or model.

3. In Table 1, the LILAC results are bolded with a caption saying that bold they means “the best case scenario”, and the main text also claims that “, LILAC is the only one to consistently increase classification accuracy and decrease the standard deviation across all datasets compared to batch learning”. However, from Table 1 it seems LILAC has neither the highest accuracy (label smoothing has overall higher accuracy), neither the lowest std. It may seem that the authors arbitrarily decided what is the best accuracy/std tradeoff which makes their method seem the best. Please define clearly the criteria for establishing the best method, and explain in what setting this criteria is a valid choice.

4. Aside from the issue mentioned above, the differences in accuracy or std in Table 1 seem minor (e.g., within 0.10% on Cifar10, and within 1% in the others). Please provide more evidence that these differences are significant.

5. How was the consistency in table 1 decided? Please provide the specific metric.

6. Regarding the choice of models and datasets, the method was only tested on image datasets. This is could be enough as a contribution, but in this case the introduction and abstract should not claim a generality that has not been tested. Similarly, the paper only considers 1 model per dataset. Does this work for other models too?

7. From Table 3, it looks like the LILAC w/o AC is actually worse than the baseline. In this case, what is the benefit of having the label introduction phase? Why not just have the AC component alone? If there is a reason, please include the results for AC alone in Table 3.

8. I find the comparison in Figure 2 potentially misleading, because I believe 1 epoch of batch training is not equal in terms of amount of training as 1 full span of the incremental phase. I believe these embeddings should be shown at convergence time.

9. In Table 4 and accompanying text, the authors conclude that “LILAC is relatively invariant to the order of the label introduction”. However, to me both random and ascending order seem actually random with respect to how this order is used. I advise the authors to try other orderings too, such as sorting them by the error of an initial training of a classifier, or other more difficulty-based orders.

---- Minor issues ----
1. In Algorithm 1, there is a missing equation (“Eqn. ??”), also I’m not sure why the first line says “Write here the result”.

2. “small batch sizes help achieve more generalizable solutions, but do not scale as well to vast computational resources as large mini-batches” → this is a bit confusing. How do large mini-batches scale better, and what is the difference between “small batch sizes” and “large mini-batches”?

---- Questions ----
Please see the major issues questions above.

**Experience Assessment:**

I have published one or two papers in this area.

**Review Assessment: Checking Correctness Of Derivations And Theory:**

N/A

**Review Assessment: Checking Correctness Of Experiments:**

I carefully checked the experiments.

**Review Assessment: Thoroughness In Paper Reading:**

I read the paper thoroughly.

---

> ### Author Response · Authors · 2019-11-12
> **Clarification of Results and Detailed Response to Reviewer's Comments**
>
> We thank the reviewer for their detailed feedback on our work. Below, we paraphrase and respond to the reviewer's comments.
>
> 1. Why should the model be forced to learn a wrong label within the dataset? Can a fictional label be used instead? If not, which of the M labels can be chosen as the fake label?
> Ans. The recursive unmasking strategy will eventually reveal all the true (data, label) pairs to the model which will remain unaffected by the use of a fake label, fictional or from within the dataset. The underlying intention of having a fixed fake label is to condense data in the Unseen set and learn strong contrastive embeddings for data in the Seen set.
> Additionally, the recursive unmasking strategy is simpler to implement when the fake label used is from within the dataset as opposed to an external value.
> There is no specification on which of the M labels in the dataset needs to be used as the fake label. We try to highlight this point by randomly ordering the labels and picking the last value in the set as the fake label, highlighted in Section 3.3 Ordering of Labels, and showing that there isn’t a specific pattern.
>
> 2. Explain the selection of hyperparameters.
> Ans. a) b (number of visible labels in the beginning) - In order to avoid extreme variations in the mini-batch size, which drastically affects performance, we fix the number of visible labels in the beginning to half the total number of labels in the dataset.
> b) m (number of labels to reveal at each step) - The selection of this parameter is discussed under Section 3.3 Injection of Label Groups. The main takeaway is that smaller groups allow for better performance. Apart from performance, this factor is also influenced by the amount of run-time available.
> c) E (number of epochs in each incremental phase) - We have included the experimental results of varying E under the Appendices. Varying E improves the mean accuracy in CIFAR-10 and STL-10 while being relatively inconclusive for CIFAR-100.
> d) T (threshold, in epochs, after which to start AC) - The selection of the threshold, in epochs, after which to start AC is closely related to the dataset being operated on. We observe that the value of T relates to how quickly the CNN can learn the dataset. Designating a point closer to the final few epochs is preferred.
> e) epsilon (Peak value of the target distribution) - The peak value used in the alternate target distribution needs to enforce a softer distribution value when compared to the one-hot vector. Section 3.3, Smoothness of Target Vector, highlights the impact of varying epsilon. The results indicate that the closer epsilon gets to 0.5, the better is the overall performance.
>
> 3. Please define clearly the criteria for establishing the best method, and explain in what setting this criterion is a valid choice.
> Ans. The criteria used to establish the best method is based on mean accuracy, standard deviation, and consistency when compared to standard batch learning (Section 3, Metrics). Within a specific model and dataset, we look for improved mean recognition accuracy and lower standard deviation when compared to batch learning. Across all the datasets and models we use consistency to indicate the general applicability of a method. Consistency is achieved when a method has improved mean accuracy and lower std. than batch learning across all selected datasets. We use mean accuracy and standard deviation as indicators for the strength and stability of the solution space learned. Through the experimental section, we highlight LILAC’s ability to improve mean accuracy, reduce std. and obtain consistency.
>
> 4. Please provide more evidence that these differences are significant.
> Ans. We perform the ANOVA 1 way test and T-test to highlight the significance of results.
>
>                                     |   CIFAR-10   |   CIFAR-100   |    STL-10     |
> Baseline vs. LILAC    |   0.45098     |    0.05517       |    0.19545    | ANOVA
> Baseline vs. LILAC    |   0.22543     |    0.027596     |    0.097717  | T-test
>
> 5. How was the consistency in table 1 decided? Please provide the specific metric.
> Ans. Consistency is a binary metric defined as follows, if a method has improved mean recognition accuracy and lower standard deviation than its corresponding batch learning baseline, across all the chosen datasets and models, then it is consistent (Section 3 Metrics).
> This is indicated by the checkmark.
>
> 6. The paper only considers 1 model per dataset. Does this work for other models too?
> Ans. Within the appendices, we provide experimental results on the HMDB51 data for the C3D model. We plan to extend LILAC to other problem domains and models in future work.

---

> > ### Author Response · Authors · 2019-11-12
> > **Clarification of Results and Detailed Response to Reviewer's Comments (contd.)**
> >
> > 7. Clarification in Table 3 for only AC method and what is the benefit of label introduction?
> > Ans. In Table 3, the row labeled Batch + AC indicates the performance of the AC in conjunction with batch learning or AC alone as mentioned in the comment. LILAC w/o AC definitely performs worse than the baseline.  However, together with the incremental label introduction, AC improves the final mean accuracy (higher) and standard deviation (lower) over both the batch learning baselines and batch + AC. The benefit of AC in LILAC is in how both phases of training function together to improve, consistently, over the batch learning baseline across all datasets.
> >
> >
> > 8. Comparison in Fig. 2 is potentially misleading. I believe these embeddings should be shown at convergence time.
> > Ans. We agree that the amount of training remains unequal between 1 epoch of batch learning and the full span of the incremental phase. However, we use Figure 2 to highlight the relative difference in the starting point of optimization when the models have access to all the ground-truth (data, label) pairs. The essence of the incremental label introduction phase is to build a strong understanding of data which can then be further optimized. At the convergence point, tSNE plots usually do not have sufficient depth to highlight the finer details and differences between the embedding spaces learned by batch learning and LILAC, given the high performance values.
> >
> > 9.  I advise the authors to try other orderings too, such as sorting them by the error of an initial training of a classifier, or other more difficulty-based orders.
> > Ans. We have updated Table 4 with results from ordering obtained from the baseline classifier, arranged in ascending order of difficulty. There isn’t a clear pattern that emerges which would necessitate identifying the label order.

---

### Official Review · AnonReviewer3 · 2019-10-24
**Official Blind Review #3**

**Rating:** 6

**Review:**

Paper summary:
This paper makes the observation that a curriculum need not depend on the difficulty of examples, as most (maybe all) prior works do. They suggest instead a curriculum based on learning one class at a time, starting with one and masking the label of all others as 'unknown' (i.e. treating them as negative examples), and unmasking classes as learning progresses. This is the "incremental labels" part. They make another observation, that label smoothing is applied to all examples regardless of difficulty, and propose an alternative "adaptive" version where labels are smoothed only for difficult examples. This is the "adaptive compensation" part.

Paper contributions:
- the two observations described above are both interesting, and methods addressing these seem like good ideas in light of the observations
 - the explanation of the method and experiments are very clear.
 - ablation and exploration studies are well-done to further investigate the proposed methods

Review summary & decision:
I like the core of this paper a lot, and recommend acceptance. It makes some insightful observations, reasonable suggestions, explains things clearly, and does reasonable experiments. The observations and proposed methods are both valuable contributions to the field. If the related work and clarity of abstract/intro are improved, along with addressing false claims and some other relatively minor things, I think this could be a very good paper, and I would be happy to increase my score.

Reasons for decision:
 - Interesting observations are made and the approaches taken are interesting and well-motivated.
 - The paper overall is well structured, easy to understand, and thorough in the experiments.
 - Some related work (detailed below) in other fields and just about curriculum learning seems to be missing. Very strange claim (which is false, as far as I know) that curriculum learning has only been used in shallow networks emphasizes that the related work is lacking.
 - The abstract and intro summary of contributions didn't do a very good job of conveying what the methods do, although they are clearly explained elsewhere (see suggestions below)
 - A lot of time is spent on detail of experiments and long, clear explanations (this is great), but makes it read a bit like a lab report. Figure 2 and the "properties of LILAC" section are nice, but could be improved with more discussion and reference to other works/fields to provide insight about _why_ LILAC works, not just _that_ it works.
 - Results seem very incremental to me (improvements over other methods are in the decimal places), and I think many people will criticize or dismiss the methods on that basis. I hesitated about this, and in the end decided that the interesting observations are the most valuable part of the paper, not pushing SOTA (while of course it's valuable to report numbers, I think our field should focus less on SOTA numbers in general).
 - Misleading results are presented for CIFAR-10; ShakeDrop is not SOTA, and no reason is provided for why the particular citations in that table are there (there are many others that could be...). I don't mind that you don't get SOTA, but I mind being misled. If I've misunderstood something here, please clarify! :)

Feedback/suggestions/nits (not necessarily part of decision assessment):
1. Briefly review some continual learning work; incremental labels are very similar to open set learning; this is worth mentioning. Would also be nice to mention anomaly detection here.
2. Briefly review negative mining (cases where this improves learning e.g. hard negative mining in text).
3. More discussion of motivation and why you think that LILAC works would be nice; connections to the above-mentioned fields could help.
4. Space could be made for the things I suggest here by decreasing spacing in the "main contributions" bullets and reducing the size of Figure 1 (the coloured tiles are very large and there's a lot of unnecessary white space. The size of text is mostly good, although the legend could also be decreased in size)
5. Incremental labels are well-explained in the dedicated section, but not until then. In the abstract, intro, and first section on "LILAC", I didn't really understand what was going on. I would suggest not using the words mask/unmask; this term is overloaded and gives the wrong connotation to me (that you don't use the predictions for the masked out classes). If you can't find a different term, I'd suggest explaining more clearly what it means to "mask" in this context in the abstract. It would also be good in the abstract to have a sentence motivating your approach (after "... learning difficult samples"). The next sentence sort of tries to do this "... starting point from which" but I think this is really unclear and makes it sound like you're learning an init.
6. Adaptive compensation is well explained in the abstract, but very confusingly in the intro summary of contributions.
7. I think it would be clearer to call it "adaptive smoothing". I know LILAC is a nice name, but so is LILAS.
8. Cite first sentence in abstract or reword
 9. "in this work, we propose.... which introduces" propose/introduce is repetitive
10. "to the best of our knowledge," [insert "all" here]
11. "curriculum learning approaches have only been tested using shallow networks" This is just false as far as I know; apart from the misleading presentation of CIFAR-10 results, addressing this is probably the most important suggestion I have. e.g. here's a whole thesis on doing CL with deep convnets that I found from a few minutes googling: http://www.diva-portal.org/smash/get/diva2:878140/FULLTEXT01.pdf
12. LILACto -> LILAC to
13. inconsistent use of italics vs. bold for emphasis in the text. Typically use italics; bold is for keywords and subtitles.
14. Mention tsne hyperparameters in an appendix
15. Conclusion is more of a summary; the first couple sentences are good but the repeating of the explanation of both methods could be moved to the abstract and/or intro (these sentences are much clearer than the ones used there), making more room to discuss future work and connections to other fields.
16. CIFAR-10 table should list method names, not just citations (they're referred to by name in the caption, which makes it hard to read the table)

 Questions:
1. What other explorative experiments could you do to investigate properties of IL and AC?
2. Did I misunderstand something about the CIFAR-10 SOTA presentation (is there some reason for the choice of numbers cited here?)

**Experience Assessment:**

I have read many papers in this area.

**Review Assessment: Checking Correctness Of Derivations And Theory:**

N/A

**Review Assessment: Checking Correctness Of Experiments:**

I carefully checked the experiments.

**Review Assessment: Thoroughness In Paper Reading:**

I read the paper thoroughly.

---

> ### Author Response · Authors · 2019-11-12
> **Clarification of CIFAR-10 results and Detailed Response to Reviewer's Comments**
>
> We thank the reviewer for their detailed feedback on our work. Below, we paraphrase and respond to the reviewer's comments.
> 1. Discussion on the motivation of LILAC would be nice. It’s links to curriculum learning and relevant fields like continual learning, anomaly detection, and hard negative mining are missing.
> Ans. LILAC was conceived to learn strong embeddings by using the recursive training strategy of incremental learning alongside unlabelled/wrongly-labeled data as hard negative examples.
> At its core, LILAC tries to learn stronger embeddings by pitting a small number of classes against the entire remaining dataset for a fixed learning interval.
> In recursively unmasking labels, the model has sufficient time to develop a strong understanding of each class by contrasting against a large and diverse set of negative examples.
> Further, when comparing the starting points when the model knows the entire dataset, optimization starts from a more well-structured point in the solution space in LILAC than in the case of random initialization (Figure 2).
> Secondarily, LILAC takes advantage of its learned embeddings by using it to identify misclassified samples and softening their target vectors to both, reduce the number of misclassifications as well as gently adapt the embedding space to incorporate them. Softening target vectors intends to make learning simpler.
> While incremental learning and continual learning seek to work on evolving data distributions with the addition of memory constraints ([1,2] and derivative works), knowledge distillation ([3,4] and similar works) or other requirements, this work is a departure into using negative mining and focused training to improve learning on a fully available dataset.
> In incremental/continual learning works, often the amount of data used to retrain the network is small compared to the original dataset while in LILAC we fully use the entire dataset, distinguished by Seen and Unseen labels. Thus, it avoids data deficient learning.
> Further, works like [5,6,7] emphasize the importance of hard negative mining, both in size and diversity, in improving learning. Although the original formulation of negative mining was based on imbalanced data, recent object detection works have highlighted its importance in contrasting and improving learning in neural networks.
> We have added this explanatory paragraph to the updated version of our draft.
> [1] iCARL: Incremental classifier and representation learning
> [2] End-to-end incremental learning
> [3] Progress & Compress: A scalable framework for continual learning
> [4] Experience replay for continual learning
> [5] Hard negative mining for metric learning based zero-shot classification
> [6] Bootstrapping visual categorization with relevant negatives
> [7] Unsupervised learning of visual representations using videos
>
> 2. The assertion that curriculum learning has only been tested on shallow networks is false.
> Ans. We apologize for the incorrect statement in our submission. We were looking at it from the perspective of top-performing approaches on CIFAR-10/100 or STL-10 and were unable to find published works that matched the criterion. We have amended the aforementioned statement to reflect this in the updated version of the paper.
>
> 3. CIFAR-10 SOTA results don't match the currently available values. Please Clarify.
> Ans. The CIFAR-10 SOTA results mentioned in Table 2 highlight performance for methods that use only simple preprocessing, random flip + random crop, no architecture search and no pretraining to highlight the impact of LILAC. LILAC was initially proposed and tested on networks that take fully image inputs and as formulated currently cannot compensate for partial inputs, be it at the node or input level.
> Given the current formulation, we removed the comparisons with cutout or random erasing data augmentation methods and selected a 3.41% error rate at 300 epochs (same as our method) mentioned in Table B1 of Shake-drop regularization (https://arxiv.org/pdf/1802.02375.pdf).
> Other methods such as GPipe, XNAS, AutoAugment, and Neural Architecture Search violate our assumptions of a simple setup (including pretrained weights on ImageNet). Hence we avoid comparing against them.
>
>
> 4. What other exploratory experiments could you do to investigate the properties of IL and AC?
> Ans. The immediate next steps for IL are to observe the impact of varying epoch sizes for each fixed interval on the embeddings learned.
> To retain the knowledge obtained at the end of IL, we plan to enforce a strong penalty when it diverges in performance, which is what happens when standard batch learning is added at the end.
> W.r.t. the AC phase, we plan to add a confidence measure in the predictions so that it can handle partial inputs.
>
> 5. All necessary grammatical changes and suggested explanations.
> Ans. We thank the reviewer for their detailed feedback and have attempted to address all the suggested changes in the updated version of the paper.

---

### Decision · Program_Chairs · 2019-12-19

**Decision:**

Reject

**Comment:**

While the reviewers appreciated the ideas presented in the paper and their novelty, there were major concerns raised about the experimental evaluation. Due to the serious doubts that the reviewers raised about the effectiveness of the proposed approach, I do not think that the paper is quite ready for publication at this time, though I would encourage the authors to revise and resubmit the work at the next opportunity.